# Why India is struggling to feed their young children? A qualitative analysis for tribal communities

Monica Lakhanpaul ![ORCID] ,[1,2] Susrita Roy,[3] Lorna Benton,[1] Marie Lall,[4] Rajesh Khanna,[3] Virendra Kumar Vijay,[5] Sanjay Sharma,[6] Logan Manikam,[7,8] Neha Santwani,[3] Hanimi Reddy,[3] Hemant Chaturvedi,[9] Shereen Allaham ![ORCID] ,[7,8] Satya Prakash Pattanaik,[6] Tol Singh,[9] Pramod Pandya,[9] Priyanka Dang,[3] Priti Parikh ![ORCID] [10]

**Correspondence to**
Professor Monica Lakhanpaul;
m.lakhanpaul@ucl.ac.uk

## ABSTRACT

**Objective** This interdisciplinary qualitative study aims to explore the health, education, engineering and environment factors impacting on feeding practices in rural India. The ultimate goal of the Participatory Approach for Nutrition in Children: Strengthening Health Education Engineering and Environment Linkages project is to identify challenges and opportunities for improvement to subsequently develop socioculturally appropriate, tailored, innovative interventions for the successful implementation of appropriate infant and young child feeding (IYCF) practices locally.

**Design** Qualitative research method, involving five phases: (1) identification of local feeding practices; (2) identification of the local needs and opportunities for children aged 6–24 months; and (3–5) analysis of the gathered qualitative data, intervention design, review and distribution.

**Setting** Nine villages in two community development blocks, that is, Ghatol and Kushalgarh, located in the Banswara district in Rajasthan, India.

**Participants** 68 participants completed semistructured interviews or focus group discussions including: mothers, grandmothers, auxiliary nurse midwife, Anganwadi worker, ASHA Sahyogini, school teachers and local elected representative.

**Phenomenon of interest** IYCF practices and the factors associated with it.

**Analysis** Thematic analysis.

**Results** Our results could be broadly categorised into two domains: (1) the current practices of IYCF and (2) the key drivers and challenges of IYCF. We explicate the complex phenomena and emergent model focusing on: mother's role and autonomy, knowledge and attitude towards feeding of young children, availability of services and resources that shape these practices set against the context of agriculture and livelihood patterns and its contribution to availability of food as well as on migration cycles thereby affecting the lives of 'left behind', and access to basic health, education and infrastructure services.

**Conclusions** This interdisciplinary and participatory study explored determinants impacting feeding practices across political, village and household environments. These results shaped the process for cocreation of our context-specific intervention package.

## STRENGTHS AND LIMITATIONS OF THIS STUDY

⇒ As an interdisciplinary study, the Participatory Approach for Nutrition in Children: Strengthening Health Education Engineering and Environment Linkages project explored a plethora of themes across the four domains of health, education, engineering and environment (HEEE) to evaluate contributing factors, current challenges and potential opportunities in infant and young child feeding (IYCF) practices in India.

⇒ The ultimate objective being to codevelop a context-specific multisectoral HEEE Nutrition Intervention Package that is socioculturally appropriate, innovative and targeted to enhance IYCF practices of 6–24 months aged children locally.

⇒ This study employed a bottom-up approach, involving a close collaboration with the local affected population, interdisciplinary teams of researchers, non-government organisations, as well as pre-existing and emergent research evidence in identifying current feeding practices and perceived barriers and in future opportunities for intervention.

⇒ Notwithstanding the scope of further research to be conducted in neighbouring areas in rural India to ensure the validity and generalisability of the findings, this study was conducted in a limited geographic scale with data gathered across two community blocks in Rajasthan.

## INTRODUCTION

The Global Nutrition Report (2018) reported that globally 150.8 million and 50.5 million children under the age of 5 years are stunted (inadequate height for age) and wasted (inadequate weight for height), respectively.[1] With 38.4% stunted children, India contributed a third of the world's burden for stunting in children under 5 years. With respect to wasting, India accounts for an even larger proportion, with 25.5 million children affected nationally, a number equivalent to half of the global burden of wasting.[2]

Conscious of the fact that adequate nutrition enhances cumulative lifelong learning capacity and adult productivity, the Government of India (GOI) recognises that improving nutrition will be of vital importance in poverty alleviation and the country's economic development in the long run.[3]

A variety of conceptual models have been used to explain the myriad and complex set of undernutrition determinants. One such model, proposed by the UNICEF,[4] illustrates the socioecological determinants influencing undernutrition. According to this model, inadequate dietary intake and infectious diseases are two key proximate factors for undernutrition alongside household food insecurity, inadequate care and feeding practices and unhealthy household environments.

Infant young child feeding (IYCF) practices entail initiation of breastfeeding in the first hour of birth, exclusive breast feeding until 6 months followed by complementary feeding after sixth months as well as optimal feeding practices essential for children until the age of 2 years. Complementary feeding includes the introduction of age-appropriate semisolid food alongside breast milk after 6 months of exclusive breast feeding.[5] Universal coverage of optimal breast feeding can prevent 13% of global deaths in children less than 5 years of age, while appropriate complementary feeding practices could result in an additional 6% reduction in under-five mortality.[6] With 80%–85% of brain growth taking place during the first 2 years of life, optimal IYCF plays an essential role also with respect to cognitive development, as reiterated by The Global Nutrition Report.[7]

When comparing India's IYCF indicators with the global and regional indicators, two important phenomena become apparent: (1) India's breastfeeding indicators (early initiation: 41.5%, exclusive breast feeding: 54.9%) are at par with the global and South Asian averages; however, they are lower than that of its immediate neighbours, Nepal (early initiation: 54.9%, exclusive breast feeding: 65.2%) and Bangladesh (early initiation: 50.8%, exclusive breast feeding: 55.3%), even though these countries have lower HDI score compared with India; and (2) India's complementary feeding indicators are lagging the global and South Asian averages and related to its neighbouring countries. While minimum dietary diversity (MDD) in India (19.9%) is similar to the South Asian average, it is lower than that of the global average (29%) and its neighbours Nepal (45%) and Bangladesh (26.6%). The major challenge is seen for minimum acceptable diet (MAD) as the rate in India (MAD 9.6%) is lower than that of the global (18%) and regional average (12%) and it is much lower than that of its neighbours Nepal (35.8%) and Bangladesh (22.8%).[8]

Complementary feeding rates declined in India between the two National Family Health Surveys of 2005–2006 and 2015–2016 from 52.6% to 42.7%. The contents of National Family Health Survey (NFHS)-5 (2019–2021) are similar to NHFS-4 and allow for comparison over time.[9] The decline was seen in nearly all the regions and states, but there was marked intracountry variation in the degree of decline. The highest decline was observed in the southern states (14%–31%), which have comparatively better performing health systems in the country, while the northern states with weaker health systems had a lower degree of decline (8%–10%). This could be attributed to the fact that the northern states already had much lower level of complementary feeding rate (during NFHS-3) compared with the southern states, so that any further decline becomes even more concerning. In the northern region, Rajasthan showed a decline of 9% in the complementary feeding rate (online supplemental file 1).

Disaggregated data from the NFHS-5 and NHFS-4[10] point to Rajasthan state in the northern region recording one of the lowest IYCF indicators. While an average of 35% children under age 3 years were breast fed within 1 hour of birth and 64% children under age 6 months exclusively breast fed, only 34% of children aged 6–8 months received complementary feeding along with breastmilk. About one-third (34%) of children age 6–23 months were fed the recommended minimum number of times per day, 9.7% had MDD and only 3.4% had access to MAD.[9 10] The MAD score in Rajasthan is the lowest score among the larger states in India.[10] Within the state, the district Banswara, which is a predominantly tribal area, has IYCF indicators worse than the state average and many of the other 32 districts.[11] Table 1 shows the IYCF indicators in India, Rajasthan and Banswara districts.

Several Indian studies have sought to unravel some key aspects of India's malnutrition scenario, addressing important determinants at three levels: (1) household (maternal time constraint, dwindling family size, mother's age and education; lack of adequate knowledge; poor

---

**Table 1** IYCF indicators in India, Rajasthan and Banswara district, NFHS-4; 2015–2016 (in %) and NFHS-5 (2019–2021)

| Indicators | NFHS-4 | | | NFHS-5 | | |
|---|---|---|---|---|---|---|
| | India | Rajasthan | Banswara | India | Rajasthan | Banswara |
| Early initiation of breast feeding | 41.5 | 28.4 | 37.8 | 41.8 | 40.7 | 33.5 |
| Exclusive breast feeding under 6 months | 54.9 | 58.2 | 57.1 | 63.7 | 70.4 | 66.2 |
| Introduction of solid, semisolid or soft foods (6–8 months) | 42.7 | 30.1 | Not available | 45.9 | 38.0 | Not available |
| Complementary feeding – minimum acceptable diet | 9.6 | 3.4 | 0.8 | 11.3 | 8.4 | 9.7 |

IYCF, infant and young child feeding.

uptake of existing nutritional services; child targeted market with wide availability and consumption of ready-to-eat marketed food items); (2) community (social and economic context; feminisation of agriculture; fragile food security/seasonal food paucity due to less focus on food crops and vegetables; dwindling livestock, especially milk producing animals; low connectivity to remote locations; migration; exposure to media); and (3) government (inadequate and unresponsive Integrated Child Development Scheme (ICDS) and healthcare system; paucity of technical knowledge among service providers regarding IYCF). Several studies over the last decade have examined varying associations between household environmental characteristics and stunting in under-five children, highlighting the need for interdisciplinary research.[9 11–16]

Our study, the Participatory Approach for Nutrition in Children: Strengthening Health, Education, Environment and Engineering Linkage (PANChSHEEEL) funded by the Medical Research Council (UK) was designed to: (1) explore health, education, engineering and environment (HEEE) factors that influenced IYCF practices and (2) develop a socioculturally appropriate, tailored, innovative and integrated cross-sector HEEE package to support optimal IYCF practices. This is central to the convergent action planning process of India's National Nutrition Mission or the Prime Minister's Overarching Scheme for Holistic Nutrition Abhiyaan that has IYCF as the first target to be monitored.

## Conceptual framework

Drawing on the review of the previous studies, the principal investigator and the coinvestigators synthesised the determinants from the literature review to frame the HEEE conceptual framework as a socioecological model (figure 1). This framework sought to synthesise the complex interplay of factors across three environments: political, village and household and the interlinkages between institutions, initiatives (schools, health services, ICDS and public distribution system) and communities. The qualitative studies that were reviewed focused on undernutrition or stunting and not specifically on IYCF and child feeding practices that have assumed crisis proportions but have received scanty attention among public health nutrition researchers in India.

The formative phase of the PANChSHEEEL study was thus framed to develop an understanding of a deep-dive interplay of the determinants of IYCF to inform the codesigning of an intervention model that can address this crisis in such contexts.

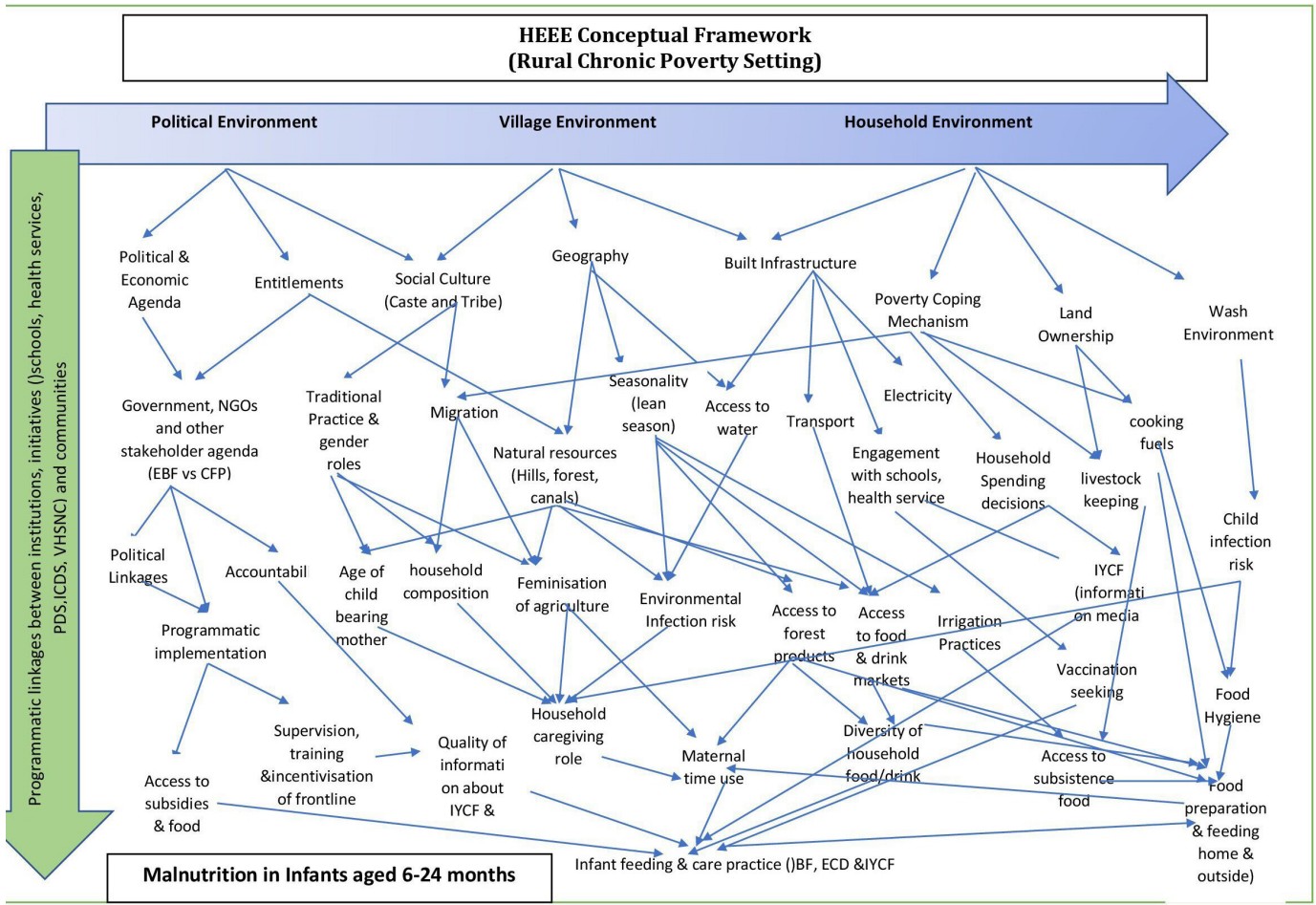

**Figure 1** HEEE conceptual framework.[20 21] HEEE, health, education, engineering and environment; ICDS, Integrated Child Development Scheme; VHSNC, Village Health Sanitation and Nutrition Committees.

This paper presents the qualitative findings of the PANChSHEEEL study that aims to identify and document local community IYCF practices including water, sanitation and hygiene (WASH) and energy practices, to identify local challenges, drivers, resources, opportunities and needs for IYCF in childred aged 6–24 months at individual, household, community and environmental level and to map the linkages between identified opportunities and challenges in order to determine how the needs identified can be addressed.

## METHODOLOGY
The larger PANCHShEEEL study used both qualitative and quantitative research methods, triangulating the two to synthesise evidence. This paper presents the qualitative component of the formative study, in line with Consolidated criteria for Reporting Qualitative research guidelines.

### Study setting
The study was conducted in nine villages in two community development blocks (hereinafter, blocks) of Banswara district in Rajasthan, India. These two blocks, Ghatol and Kushalgarh, were purposively chosen to represent district diversity. The Ghatol Block is located in the 'command area'

of the Mahi River Dam; all villages are irrigated by canals and have multiple crops yearly. In contrast, Kushalgarh is dry and semiarid with poor irrigation; consequently, most villages are mono-crop areas. The village selection process is represented in the following consort diagram (figure 2).

### Patients and public involvement and engagement
In line with the PANChSHEEEL participatory approach, considerable efforts were placed on Patient and Public Involvement and Engagement. Suitable individuals and representatives from both the Ghatol and Kushalgarh Block were identified and subsequently engaged in all steps of the study, that is, protocol development, study design and results dissemination.

The Core Research Team was further supported by community researchers from the two study blocks who aided in steering the project phases locally. With the input from 'Save the Children' Rajasthan, two well-suited individuals were identified who had been proactive collaborators in previous projects by 'Save the Children' India. They were mainly involved in selecting suitable candidates who could engage actively in project activities, as well as coordinating with the community champions, who were crucial in establishing networks with potential study participants among community members.

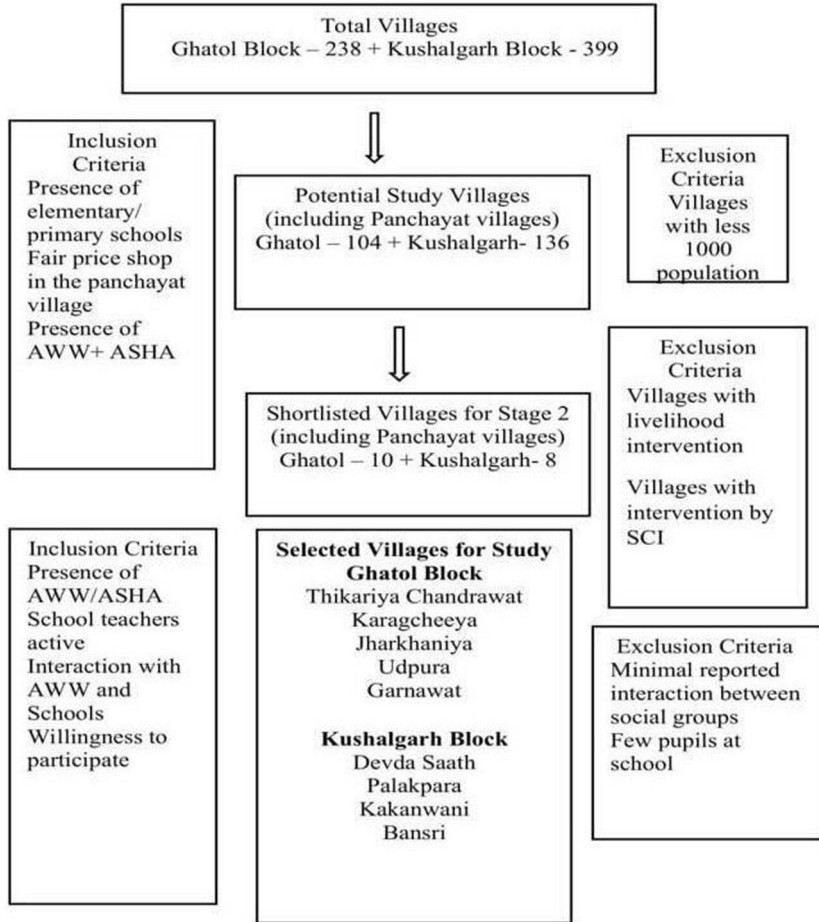

**Figure 2** Consort diagram for selection of study villages.[21] AWW, Anganwadi worker.

Relevant stakeholders were identified and selected in a stakeholder mapping workshop as part of the formative study phase. Through snowballing, that is, a recruitment method where existing participants are involved in the identification and recruitment of future participants using their social networks, the research team remained open to potential new stakeholder(s) identified during the data collection process. The recruitment of study participants was based on willingness to participate and availability.

### Respondent selection

The respondents included stakeholders who play a critical role in feeding and caring for young children. There are three cadres of frontline workers at the village level in Rajasthan: (1) the auxiliary nurse midwife (ANM), the multipurpose female health worker; (2) the Anganwadi worker (AWW), the cadre of the ICDS who runs the Anganwadi Centre (AWC) at each village; and (3) the Accredited Social Health Activists (ASHA) Sahyogini (AS), the community health worker (ASHA in other states). One ANM was selected from each of the subcentres in the study villages; in case of subcentres with two ANMs, the senior ANM was selected. Owing to the size of the village, generally one AWW or AS was chosen, but in the case where the village had more than one AWW or AS, then both were interviewed as they represented different geographic and social strata. One school from each village was chosen, and the principal/head or the most senior teacher was interviewed. The ward or local sarpanch of each village was also interviewed. At household level, mothers and grandmothers who were literate and conversational (with at least one child aged 6–24 months) were identified and selected based on their availability and willingness to participate either during the household survey or by frontline health workers (FHWs).

### Data collection

Key informant interviews (KIIs) and focus group discussions (FGDs) were used for collecting primary data. Semistructured open-ended interview guides were used to understand respondent knowledge and feeding practices perceptions. FGDs were conducted with mothers and grandmothers. The date, time and meeting place were decided in consultation with AWW and AS. FGD guides included topics on maternal time use, household care giving roles, breast feeding and complementary feeding, hygiene, play and communication. An approximate of five to nine questions were asked under each topic, and the data collected were classified into four major themes (health and nutrition, education, WASH and cooking Fuels). Participants were encouraged to share their frank opinions, irrespective of the views of others in the group, in order to elicit insights about IYCF issues (table 2).

FGDs were conducted by community researchers from the same blocks or adjoining block who were able to communicate fluently in the local dialect, *Wagdi*. They were trained on FGD techniques by the principal

**Table 2** Respondents categories and methods

| Category of respondent | Method used | Numbers |
|---|---|---|
| Mothers and grandmothers | FGD | 17 |
| ANM | KII | 7 |
| AWW | KII | 13 |
| ASHA Sahyogini | KII | 13 |
| School teacher | KII | 9 |
| Local elected representative | KII | 9 |

ANM, auxiliary nurse midwife; AWW, Anganwadi worker; FGD, focus group discussion; KII, key informant interview.

investigator and coinvestigators that included principles and methods of qualitative research with a special focus on interview and group discussions and an outline of data analysis. Hands-on support was provided by other team members who were well versed with the research methods and local contexts. The duration of the interviews ranged between 25 and 40 min and the FGDs ranged between 4 and 75 min.

We relied on a hybrid data saturation approach: achieving both: (1) a priori thematic saturation (the degree to which identified codes or themes are exemplified in the data) that informed sampling strategy and (2) inductive thematic saturation (the emergence of new codes or themes) that informed data analysis.[17]

Field notes based on observations played a crucial role in data collection and analysis. Digital recorders were used to record all KIIs and FGDs. Written consent was taken from key informants while for FGDs verbal consent was sought and formally recorded. Audio data were then transcribed (from local language to Hindi) and translated (from Hindi to English) with back translations done for quality assurance. Both audio data and text were anonymised to maintain confidentiality and securely stored. A pilot round was conducted in January 2018 following which formal data collection continued for 4 months. Caution was maintained regarding selection bias of FGD participants due to the active role of FHWs in service delivery. FGDs were organised at ICDS or AWCs at each village with care taken to ensure that residents residing far from AWCs were not excluded.

### Topic guide

Through the means of a literature review, expert advice and pilot testing with different population subsets, a thematic guide was designed prior to the qualitative data collection process. This guide facilitated the conductance of the KIIs and FDGs, allowing the topic of IYCF to be explored across the four major dimensions: (1) health and nutrition, (2) education, (3) WASH and (4) energy to capture current practices, barriers and possible opportunities to foster positive change across all four areas.

## Data processing and analysis

The research team undertook daily meetings to reflect on their key observations and data collected from field notes. Data were transcribed and translated to bring out common major themes based on commonality and differences. Data generated from all sources were subjected to triangulation of mixed qualitative and quantitative methodology[18] and linked thematically for key findings. The qualitative data analysis software, IQDAS, was used for the data analysis process. The data were coded and analysed jointly by the manuscript authors (coinvestigators) to ensure consistency and reliability in interpretation. Codes were analysed across axial and selective codes across respondent categories and sites (villages and blocks). Triangulation was performed across methods and respondents. Results from KIIs and FGDs were compared and assessed for similarities and differences across respondent categories. Findings from villages within a block conveyed similar interpretations and hence reinforced data validity. The investigator group reached a consensus to merge village and block level findings, except in those themes or subthemes where they presented novel findings. Data analysis was first completed independently by interviewers followed by a series of joint sessions between interviewers and coinvestigators.

## Respondent validation

A series of meetings were conducted with representatives from all respondent categories from the same villages to validate the data. Within the qualitative data sets, all steps of a thematic analysis, that is: (1) identifying, (2) analysing and (3) interpreting patterns of meaning, that is, themes, were employed. Respondent validation was facilitated by community researchers in September 2018 and closely overseen by local Save the Children staff. This enabled us to refine the emergent themes and model to its final shape.

## Quality assurance

The interdisciplinary team comprised of experts in public health nutrition, epidemiology, social sciences, paediatrics, education, and civil and environmental engineering. A dedicated team of coinvestigators supervised training, data collection, transcription and translation, data management, and analysis. Field teams were oriented with the study protocol, tools and research techniques, including hands-on exercises, before actual data collection began.

## RESULTS

A total of 68 participants were recruited (table 2). The information was obtained from FGDs with 17 mothers and paternal grandmothers and 51 key respondents across nine villages. More than half of the mothers were aged between 18 to 26 years, and majority of them (approximately 92%) had either one or two children under 5 years in their household. Sixty-one per cent of the women including both mothers and grandmothers were illiterate. Of the rest who attended formal schooling, there was roughly a 10% dropout rate because they got married early or participation in household work. More than 90% of the women interviewed were involved in the agriculture industry either in the form of farming or livestock rearing. Based on the present study that defines time spent in tending to farming and livestock as 'market work', women spent a greater amount of time in activities of farming and tending livestock.

Deriving from the study objectives, this section is divided into two parts. The first section provides an in-depth description of IYCF practices. The second section will explain the factors (drivers and challenges) associated with it. Online supplemental file 2 illustrates the themes identified and example quotes from focus groups and interviews with respondents. Figure 3 illustrates the current practices of IYCF.

## IYCF practices
### Core theme 1: partial breast feeding is a common practice in first 6 months

Most mothers and grandmothers reported that 90% of the births take place in hospitals due to the presence of trained staff who are equipped to handle medical emergencies and breast feeding was initiated within the first 2–3 hours of childbirth in hospital. Of the few women who had given birth at home, they also confirmed that breast feeding starts as soon after birth as possible. Delay in breastfeeding was reported mainly due to '*secretion of milk not starting*' (grandmothers FGD, Kushalgarh).

> If the baby is delivered at home breastfeeding is starts immediately after birth. (AS, Kushalgarh)

Giving colostrum was considered mandatory among a majority of respondents, with only a minority of grandmothers in Kushalgarh expressing negative views about colostrum as 'dirty milk'. All respondents denied giving any prelacteals like honey to the children though these practices were reportedly followed in cases of home births, although very few.

Most mothers in Ghatol reported that they exclusively breast fed for 4–6 months. In Kushalgarh, a majority of respondents breast fed until the third month. Breastfeeding frequency in the first 6 months was reported to be 'on cue', as and when the child cried.

> Mothers of children who are on breast milk work inside the village, so they can be summoned home whenever the child cries. (Mothers, Ghatol)

Mothers who resumed their agricultural work after 3 months reported coming home from work every 2 hours for breast feeding. On further prompting, it was revealed that, at times, water was given to children below 6 months, especially during summer months due to the belief that the child gets thirsty. Dissolving biscuits in milk or water and feeding it to children in case they cried or the mother was not available appeared to occur on occasion,

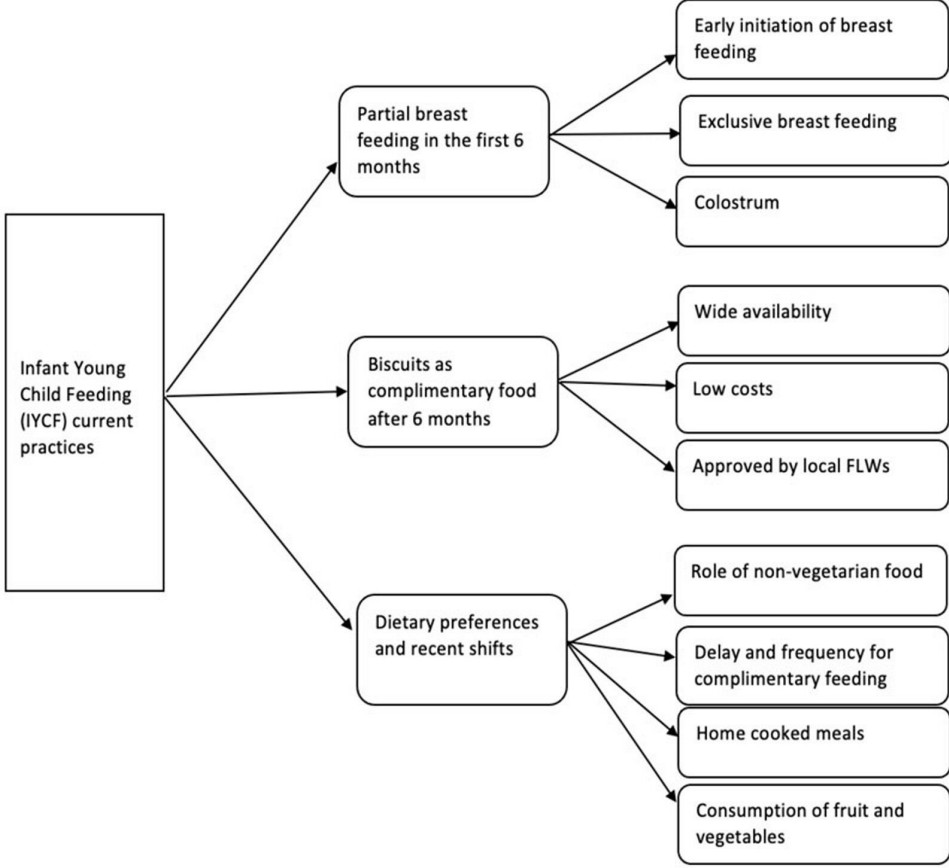

**Figure 3** IYCF – current practices. FLWs: Frontline Workers.

thus compromising exclusive breast feeding. This practice was also promoted by FHWs. Animal milk, especially goat milk, was given when the mother was unavailable or when the mother's milk amount was perceived to be inadequate.

> In the summer season the child becomes thirsty, so water is given to the child. We also recommend giving water sometimes. (AWW, Ghatol)

### Core theme 2: biscuits given as complementary food after 6 months

Consumption of biscuits as convenience foods increased after 6 months in both blocks. Soaked in water or tea, they were used as a semisolid food for local children. Two reasons were cited for this practice: (1) easy availability and low costs (5 INR or US$0.7 per packet) and (2) children liked the sweet taste, hence making it easy to feed.

> Children like biscuits and it is easy to feed them. (Grandmothers FGD, Kushalgarh)

> This is the cheapest option and is easily available in the village. (Mothers FGD, Ghatol)

These were given in the morning as the first meal of the day, and/or between meals when children cried. FHWs commonly approved of this practice.

> We tell the mothers that they can give biscuits to the child, at least it will keep them full. [ANM, Kushalgarh]

Biscuits were often employed as a method to *pacify* the child when the mother or other caregivers were busy in their daily chores. The number of biscuits given per day varied from 5 to 10 per day.

### Core theme 3: dietary preferences and recent shifts

On average, complementary feeding was commenced around 7–8 months. The main reason of delayed complementary feeding was the child's 'lack of interest' in eating semisolid foods, marked by prolonged feeding time, crying and also fidgeting while eating. A majority of participants reported to be confident regarding appropriate complementary feeding.

> The baby starts drinking milk (dairy) after 6 months. We know what to give and what not to. (Mothers FGD, Ghatol)

The foods included pulses (in watery consistency), *khichdi* (savoury rice and pulse gruel), *dalia* (porridge with ground wheat with milk or water, mostly sweet) or small pieces of roti dipped in milk. The use of commercial baby food was nominal. FHWs reported that children were fed four to five times a day on average; mothers and grandmothers, however, reported demand feeding to

be common practice, that is, feeding when the children cried.

> Whenever the children get hungry we feed them. (Mothers FGD, Kushalgarh)

Most mothers from both blocks reported that, because children ate in small quantities, it was difficult to cook separately for them every day. Neither FHWs nor the mothers had clarity about the frequency and quantity to be fed to children. Portion size varied depending on the child's 'interest'. All mothers confirmed that they received packets of take-home ration regularly but had little knowledge of the proper recipe.

Consumption of non-vegetarian food was dwindling while those who consumed animal products, for example, meat and eggs, did so discretely. This was attributed to religious reasons, locally termed the '*bhagat*' (disciple) culture.

> This is a recent trend. Hawan (worshipping with holy fire) was conducted in the village after which every family has stopped eating. No question of giving it to children. (Ward Panch, Ghatol)

This practice was more prevalent in Ghatol, where more respondents ascribed to these beliefs.

In Kushalgarh, hens were reared in most households, consequently resulting in a higher availability and consumption of non-vegetarian foods. However, due to religious values, respondents often denied its consumption.

> This is made only if there is a guest at home or there is any occasion. Chicken is mostly eaten (on such occasions). (Teacher, Kushalgarh)

Responses from both blocks confirmed that children under 5 years were rarely given non-vegetarian food, and if so, only in small quantities.

Consumption of fruits, though unanimously considered to be beneficial, was rare on account of availability and cost. Fruits are generally available in the markets located far away from the villages, more so in Kushalgarh. Purchase was possible only when a family member visited these markets.

> Fruits are not very common in daily diet. Only when the parents go to the market they get fruits. (Teacher, Kushalgarh)

Vegetables consumption also comprised '*potatoes, tomatoes and onions almost daily*' (ANM, Ghatol). There was less consumption of green leafy vegetables, for children, even though it was readily available.

### Key drivers and challenges

IYCF practices described in the previous sections are attributable to contextual issues emerging at the household, community and government level. Differences evident between Ghatol and Kushalgarh include cropping patterns, market access and fruit and vegetable availability. These differences aside, there were similarities in the drivers of IYCF practices between these two blocks. Figure 4 presents the connecting subthemes of the key drivers and challenges of IYCF practices.

### Core theme 4: time-constrained mother and role of other family members

Time constraints among mothers emerged as a critical determinant accounting for lack of exclusive breast feeding (infants below 6 months) and feeding (home cooked food) at regular intervals (to older children).

> The mother cannot sit at home taking care of the child. Who will look after the animals, get water or cook? They also have to do agricultural work. (Grandmothers FGD, Kushalgarh)

All mothers, while being primary caregivers, reported that they were guided and influenced by grandmothers, other family members and FHWs. Mothers stayed at or worked in close proximity to their homes until the child was about 2 months old. During this time, they continued with cooking and other household chores, as well as outdoor work, that is, firewood and water collection. Mothers from Ghatol reported that they did not go out for work outside the villages. Occasionally, mothers took up daily work in nearby factories, but only once their child was at least 1 year of age. Women in Kushalgarh migrated to nearby towns and districts as a family, once children were above 6 months.

> When my daughter-in-law works outside, then we grandmothers take care of the children. (Grandmothers FGD, Ghatol)

Teachers reported that also elder siblings, especially girls, were often absent from school to take care of younger siblings, especially in families where both parents had migrated (seasonally) or if the primary caregiver was unavailable.

### Core theme 5: livelihood challenges in a chronic poverty setting

Agriculture is the main source of livelihood and revenue for families in both blocks, despite clear differences in irrigation facilities and cropping pattern in Ghatol and Kushalgarh.

> Most people here are in agriculture. Both males and females of all families take part in agriculture. (School Teacher, Kushalgarh)

Women in all study villages were engaged in agriculture and livestock farming throughout the year; men took part in agriculture only during sowing and harvesting seasons. Some women were engaged under the National Rural Employment Guarantee Scheme though employing mothers with children aged less than 2 years was not common practice. Local wage labour was reported to be a common source of income for men residing in Ghatol villages due to the proximity to the district town and a cloth mill. Circular migration to urban areas of

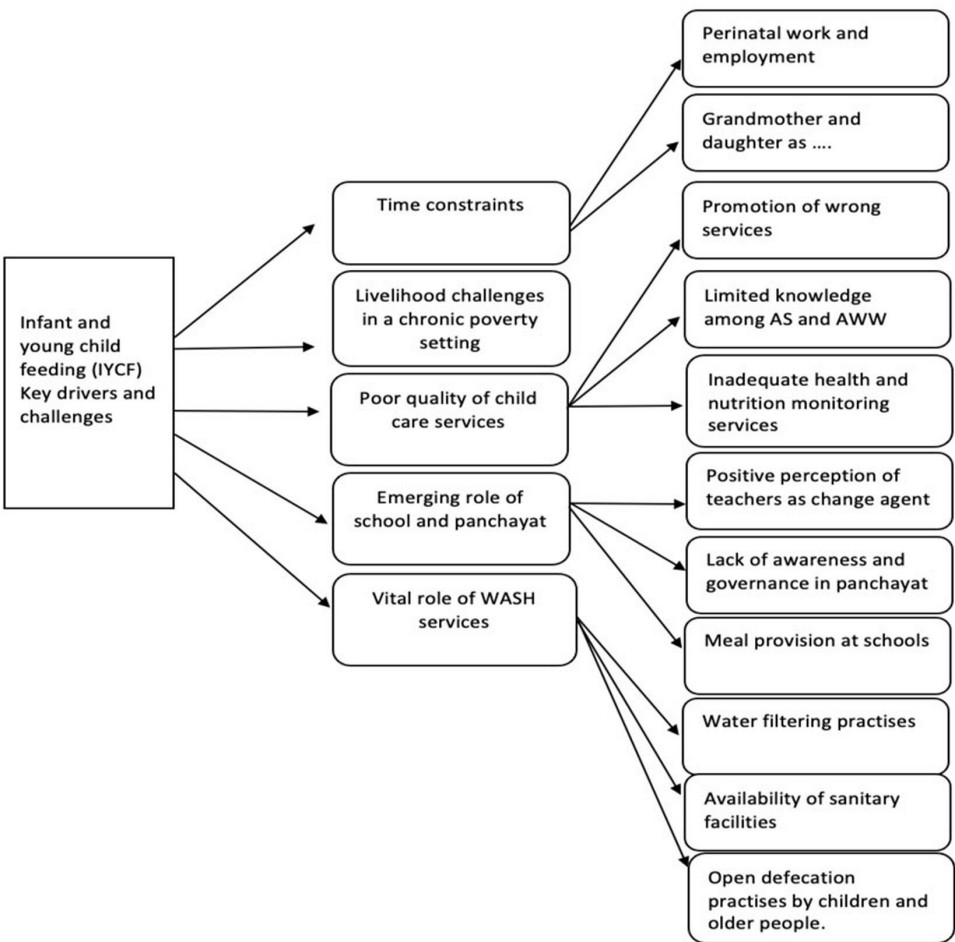

**Figure 4** IYCF – key drivers and challenges: connecting subthemes. AWW, Anganwadi worker; AS, ASHA Sahyogini; WASH, water, sanitation and hygiene.

the adjacent states of Madhya Pradesh and Gujarat was common among men in Kushalgarh.

> Some men from the village work in Mayur mills (textile mill near Banswara). Some also work as wage labour, but only during lean seasons. (School Teacher, Ghatol)

### Core theme 6: childcare services fails to deliver on most counts

Proactive steps are being taken by government programmes to address IYCF-related challenges. Breastfeeding information was mostly provided at childbirth facilities, and these practices were much better compared with complementary feeding. The overwhelming evidence from all the villages was that level of knowledge and counselling skills were poor among AWW and AS. They were not aware of malnutrition indicators and appropriate complementary foods, often promoting the introduction of animal milk, semisolid or solid food into a baby's diet before the age of 6 months. Infant weight was recorded in some cases.

> We take weight, but currently the weighing machine is not working. (AWW, Kushalgarh)

In many villages, several (non-health) key informants reported that AWWs mostly catered to community need staying close to the AWC. The supervisory cadres rarely visited the villages.

> Neither LS (Female Supervisor) nor the CDPO (Child Development Project Officer) visits our centre. We only go to the sector meetings where they ask us about the number of pregnant women registered, children born and packets of THR distributed. (AWW, Kushalgarh)

Ghatol received more support from the district level than Kushalgarh owing to its proximity to the headquarters; this was further complemented by the presence of NGOs and other civil society actors in Ghatol.

### Core theme 7: emerging role of schools and panchayats in health and nutrition programmes

School teachers were an important resource for local communities; the schools functioned regularly and efficiently with teachers attending school daily. Mid-day meals were provided. With male members mostly working outside the village (either daily wage or migration), parents' participation in these meetings were occasional.

According to school teachers (mostly male), fathers interacted with them with few mothers participating. The state has recently formulated at policy for colocation of AWC at school premises.

> The AWC is in the school premises; we can always keep a check whether children are getting their meals. The AWW and the other staff also come in time and regularly open the centre. (School teacher, Kushalgarh)

Many teachers reported taking a personal interest in visiting and monitoring the functioning of AWCs; however, due to their own busy schedule and excessive workload (in part due to staff shortage), they were not always proactive.

> We are already overburdened with the work of the school. It is not possible to check the functioning of the AWC. (School teacher, Ghatol)

All the ward Panches (elected Panchayat (local self-government) member) were unaware of the Village Health Sanitation and Nutrition Committees and played no role in this critical village level platform for promoting health and nutrition of mothers and children.

### Core theme 8: WASH practices can play a vital role

Caregivers were unanimous in their perception that illnesses such as diarrhoea, fever or cough represented one contributing factor of inappropriate nutrition among children. There was little awareness that these diseases may be partially linked to WASH deficiencies. Hand pumps were the only source of drinking water in both blocks. Unlike in Ghatol where a majority of hand pumps have water throughout the year, some sources in Kushalgarh dry up '*due to scarcity of water*' (ANM, Kushalgarh]) during the summer months. The only measure for drinking water treatment was using a cloth for filtering the water while filling; this practice (as reported by the FHWs) was more common in Kushalgarh than in Ghatol. In several villages of Kushalgarh, a local reverse osmosis plant using solar panels was set up under a new scheme by the Public Health and Engineering department.

Mothers, grandmothers and other key informants reported that communities in Ghatol were more willing to construct sanitary facilities than those in Kushalgarh, even though there is a national sanitation programme that incentivises toilet construction at household level.

> Those who constructed the toilets have not got their due payments. I have been following up with the Sarpanch's office regularly, but there is no response. This discourages those who have not yet constructed. (Ward Panch, Kushalgarh)

In both blocks, however, a majority of households had access to functional toilets.

> The government has constructed toilets in every household. Whatever facilities the government has provided us, I have provided in this village. (Ward Panch, Ghatol)

Most members reported using toilets, with the exception of a few elders who prefer open defecation. None of the under-two children defecated in toilets. Small children defecated on pieces of cloth, which were subsequently disposed of in open fields near the house. The older children defecated in the open, near their homes, except some in Ghatol who were reported to use toilets.

Awareness on the importance of hand washing was high in all study villages. Hand washing with water and soap before cooking and eating was unanimously reported in Ghatol, but less so in Kushalgarh due to water scarcity.

## DISCUSSION

IYCF practices can be categorised as optimal and suboptimal. Optimal practices such as early breastfeeding initiation, reduction in prelacteal use, mothers being encouraged to exclusively breast feed the child (until 6 months) and continued breast feeding until 2 years were reported in our study villages. Suboptimal practices were more widespread and included giving water in the first 6 months, feeding biscuits from the fourth month and complementary feeds lacking in quantity, frequency and diversity. The emergent model bears out the interactions of the socioecological elements framed in our conceptual framework (figure 5).

It is in this backdrop that we sought to identify factors that influence and shape these existing practices and then design interventions that are able to address barriers and promote improvement. The evidence presented in this paper forms the basis of the socioculturally appropriate, tailored, integrated and interdisciplinary interventions developed as part of the PANChSHEEEL project drawing on the sectors of: (1) health and nutrition, (2) education and (3) engineering and environment.

Most of the facilitators and barriers were in the realm of knowledge and skills. The two most common barriers across most households were lack of appropriate knowledge of mothers and other caregivers about: (1) complementary feeding practices (frequency, quantity and quality) and (2) recipe for cooking take home ration (supplementary nutrition) distributed by the AWC. The knowledge gaps also resonated with incomplete information that was provided by the FHWs; this was also one of the barriers to promoting positive IYCF practices. Two such (highly prevalent) practices were giving water to babies less than 6 months and feeding biscuits. We also noted the divergence between perceptions of mothers and grandmothers. This was largely on account of mothers caring and feeding the children more actively in contrast to grandmothers who looked after children largely when mothers were away for work and were not often the final decision makers of care and feeding practices.

Resource constraints for mothers emerged as another significant factor. Lack of mother's time to continue

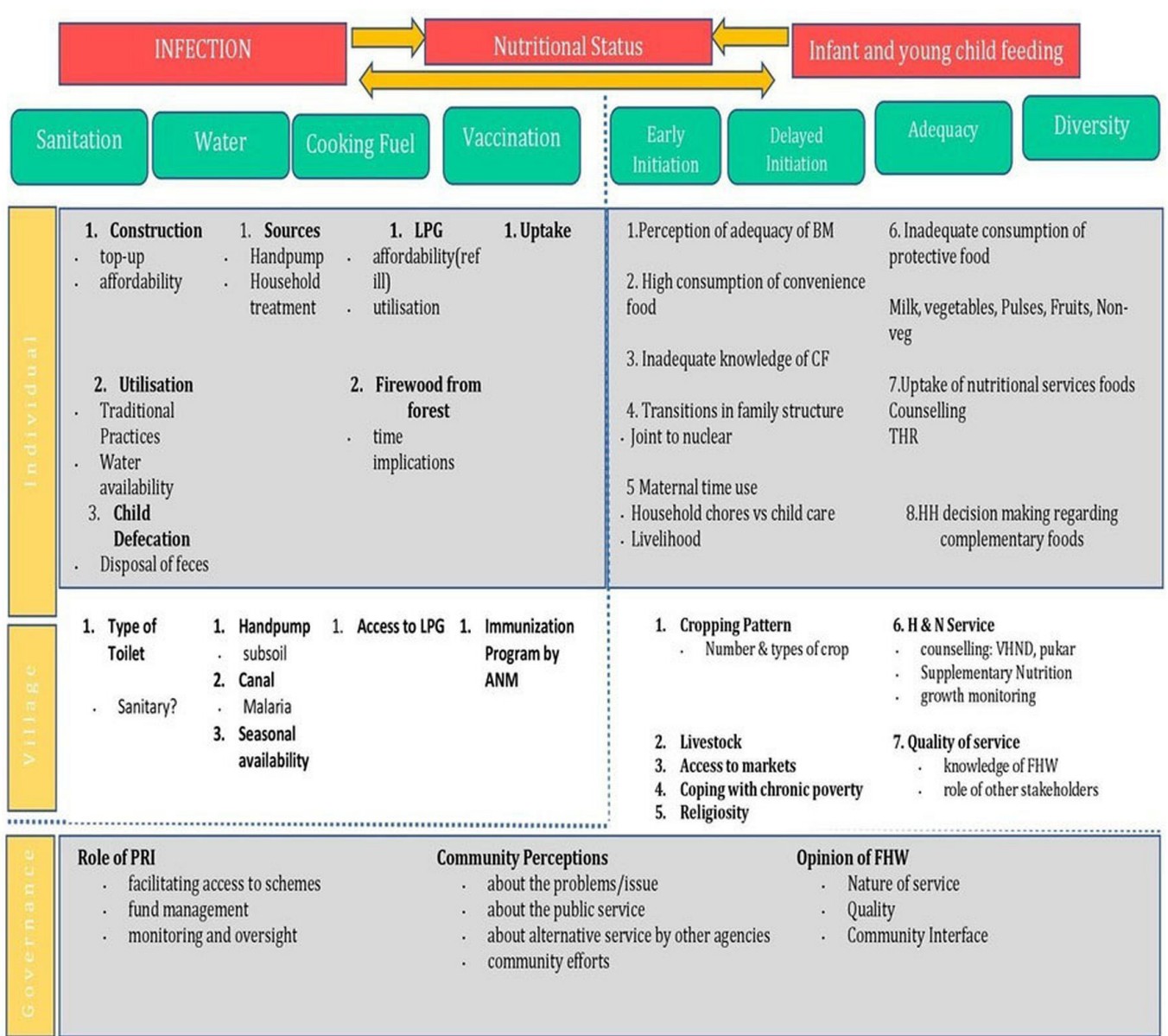

**Figure 5** Emergent model. ANM, auxiliary nurse midwife; FHW, frontline health worker; THR, take-home ration; PRI, Panchayati Raj Institutions; HH, Household; LPG, Liquefied Petroleum Gas; VHND, Village Health and Nutrition Day; CF, Complimentary Feeding; BM, Breast Milk.

exclusive breast feeding as well as dedicating time for separate cooking and feeding was one of the most common barriers to positive IYCF practice. As male members travelled to nearby towns for daily wage labour or migrated to nearby states (owing to lack of employment opportunities in the villages), women were compelled to dedicate more time to agricultural work. The poor environmental conditions and limited access to basic water and sanitation services also contributed to the time and resource constraints faced by women.

Little or no availability of animal milk resulted in children between 6 months to 2 years hardly receiving any milk in their diet. Limited use of vegetables and fruits was due lack of markets, inadequate resources to cultivate and high cost. As a result, children's diets were seldom rich in iron and vitamins. The absence of adequate resources was

not merely a household level issue, but also rooted in the local cultural and religious contexts. The remaining two determinant categories, motivation and attitude, were mostly linked to either knowledge or resources.

Delineation of these factors helped in identifying key 'actors' who could bring about improvement in IYCF practices. To ensure continued support to mothers and other caregivers on positive IYCF practices, the knowledge and skills of FHWs, especially AS and AWW, need to be upgraded. Finally, there is a need for garnering community support for these children for which the role of important members like school teachers and elected representative was also identified as crucial.

This study was also subjected to certain logistical limitations and research related concerns. The language was a barrier to be addressed, everyone knew Hindi, but only

two people in the team were versed with Wagdi, the local dialect. This resulted in challenges for translation and transcription of the data from the discussions. Moreover, the availability of the functionaries and the key stakeholders including the teachers at the schools, local official representatives were limited. This also added to the time constraint. All data collected were based on the observation and interactions held in a limited time of 2 hours per village. Sometimes the recall of information was difficult for participants; this was seen when the exact dates of births could not be retrieved resulting in calculation of the child's age based on estimated information on their month and year of birth. Besides, there were challenges to encourage women to participate in the interviews. They either did not feel empowered to contribute or were busy with household chores. For the same reason, availability of mothers was limited considering their demand for domestic work is a lot more. To encourage them there were multiple attempts to organise FGDs and KIIs. The researchers had to spend a significant amount of time with the caregivers to build the space where some of them could freely and confidently contribute to the discussion. On top of this, the decision to include mothers who have children at the age of less than 2 years old led to a varied number of eligible mothers from 35 households to 68 households in some villages, which was not align with the initial plan to cover 50 households per village. Lastly, there might be a selection bias of the participants of the FGDs due to the active role of AWW and ASHA workers in the selection of mothers, which implied that only those with whom they have a good rapport would be identified. The prior training about the process could also influence the information given by some of the mothers. Another reason for the selection bias could also be that most of the FDGs were held in the AWC, as a result of which only the residents near the AWC could take part excluding the ones that are far away. We acknowledged the limitations and aimed to find ways to overcome the challenges we faced.

This study, and the urgent need for IYCF interventions, still holds a significance in the light of the recently released Global Hunger Index (GHI) 2019 data.[19] India's GHI score declined from 38.8 in 2000 (rank: 83 out of 113 countries) to 30.3 in 2019 (rank: 102 out of 117 countries), placing the country in the 'serious' category. Achieving internationally agreed targets on stunting and wasting in under-five children is key to achieving Sustainable Development Goals (SDGs). The complex set of factors and barriers to appropriate IYCF practices—captured in the HEEE conceptual framework and the qualitative data—pose a challenge in achieving the SDGs. While the GOI has designed and implemented several programmes to address these barriers, they reported limited success primarily because of the nature of isolated interventions as opposed to much needed multifaceted approach. The task at hand is therefore a well-designed, locally feasible, multisectoral interventions across health and nutrition, education

as well as water and sanitation. Through this study, we demonstrate the need for integrated and cross-sectoral research to comprehensively shape the feeding practices of children, thereby creating a scope for coordinating the ongoing interventions in a manner to achieve the desired target (the subject of another paper).

**Author affiliations**
¹Department of Population, Policy and Practice, UCL Great Ormond Street Institute of Child Health, University College London, London, UK
²Whittington Health NHS Trust, London, UK
³Save The Children, Gurugram, India
⁴Institute of Education, University College London, London, UK
⁵Indian Institute of Technology Delhi, New Delhi, India
⁶Save the Children, Rajasthan State Programme Office, Jaipur, India
⁷Department of Epidemiology and Public Health, University College London Institute of Epidemiology and Health Care, London, UK
⁸Aceso Global Health Consultants, London, UK
⁹Save the Children, Rajasthan, India
¹⁰Engineering for International Development Centre, Bartlett School of Sustainable Construction, University College London Faculty of the Built Environment, London, UK

**Acknowledgements** Participatory Approach for Nutrition in Children: Strengthening Health, Education, Environment and Engineering Linkage (PANChSHEEEL) gratefully acknowledges the input from our community champions and field team, which made this project possible. We would like to thank MRC for funding the PANChSHEEEL project. We would also like to thank Isabel-Catherine Demel, a medical student who helped out in formatting the paper, and Shyantani Saha, a research assistant at University College London who helped in incorporating the reviews and developing the manuscript. The opinions in this paper solely reflect the views of the authors and not of organisations who have financially supported this research. All data used in this research are property of the authors. The data that support the findings of this study are available from the corresponding author, MoL, on reasonable request. MoL was supported by the National Institute for Health Research (NIHR) Collaboration for Leadership in Applied Health Research and Care North Thames at Bart's Health NHS Trust. This work is supported by the NIHR GOSH BRC. LM and SA are funded via an NIHR Advanced Fellowship (Ref: NIHR300020).

**Contributors** ML, PriP, RK and SS conceived the original concept of the study and designed the research methodology. SR, LM, PriP, RK, HC, SS, SPP, TS, HR and PraP carried out the interviews, analysed the data and wrote the paper. MoL, LM, PriP, SR, HC, NS, SS, SPP, LB, VKV, PD and RK validate the study and revised the manuscript critically for important intellectual content. SA contributed to the manuscript writing, edited the final manuscript and prepared for submission. ML is the guarantor and had primary responsibility for the final content. All authors read and contributed to the reviewing the analysis of the data, the designing of the manuscript and the approval of the final manuscript.

**Funding** This work was also supported by the Global Challenges Research Fund and funded by the Medical Research Council (MRC), Arts and Humanities Research Council (AHRC), Biotechnology and Biological Sciences Research Council (BBSRC), Economic and Social Research Council (ESRC) and Natural Environment Research Council (NERC) (grant number: MR/P024114/1).

**Competing interests** None declared.

**Patient and public involvement** Patients and/or the public were involved in the design, or conduct, or reporting, or dissemination plans of this research. Refer to the Methods section for further details.

**Patient consent for publication** Not applicable.

**Ethics approval** ]This study was conducted according to the guidelines laid down in the Declaration of Helsinki, and all procedures involving research study participants were approved by the UCL ethics (Ethics ID 4032/002) and Sigma IRB (10025/IRB/D/17-18).

**Provenance and peer review** Not commissioned; externally peer reviewed.

**Data availability statement** Data are available on reasonable request. The data of this study is available from the corresponding author on reasonable request.

**ORCID iDs**
Monica Lakhanpaul http://orcid.org/0000-0002-9855-2043
Shereen Allaham http://orcid.org/0000-0003-0275-3228
Priti Parikh http://orcid.org/0000-0002-1086-4190

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
