## [Reviewer comments · BMJ Open]

ARTICLE DETAILS

TITLE (PROVISIONAL)	Why India is struggling to feed their young children? A Qualitative analysis for tribal communities
AUTHORS	Lakhanpaul, Monica; Roy, Susrita; Benton, Lorna; Lall, Marie; Khanna, Rajesh; Vijay, Virendra; Sharma, Sanjay; Manikam, Logan; Santwani, Neha; Reddy, Hanimi; Chaturvedi, Hemant; Allaham, Shereen; Pattanaik, Satya Prakash; Singh, Tol; Pandya, Pramod; Dang, Priyanka; Parikh, Priti

VERSION 1 – REVIEW

REVIEWER	Mamidi, Raja Sriswan Indian Council of Medical Research, Clinical Epidemiology
REVIEW RETURNED	09-Jul-2021

GENERAL COMMENTS	Please remove the tables from introduction to supplementary section as it dilutes the purpose of the objective. The concept of PanchSHEEL needs to be elaborated more in the introduction/methodology/discussion. For example it is unclear from the text what is meant by "engineering". How is this framework different from the globally accepted UNICEF framework of malnutrition.
---

REVIEWER	Flores-Quispe, Maria del Pilar Universidade Federal de Pelotas, Medicina Social
REVIEW RETURNED	18-Aug-2021

GENERAL COMMENTS	My comments are listed below and are made by manuscript section. Specific comments: Title: • Ok Abstract: • The line 0 and 1, the aim, could be after the line 7 (IYCF (Infant and Young Child Feeding) practices locally.) Introduction: • In line 9, I suggest to specify that the information about stunting and wasting is for children under-five • In line 14 and 15, according to the reference number 2, the information about India is for the year 2007?, It could be possible to bring information of recent years, as the authors have made for the line 9 • In the fourth paragraph, it could be better to describe, in a brief way, the differences between India and the others countries in
--

South Asia instead the table 1, because in the introduction the authors should present a general view what is happening about the issue.

- I could suggest writing about the breastfeeding indicators and compare to Global and Sri Lanka data. Although, Sri Lanka is a country smaller than India and has Human Development Index (HDI) higher than India, so, the authors perhaps could compare with other countries with characteristics similar to India.
- Regarding to complementary feeding indicators, the authors could write comparing India with global and Bangladesh data, which has a similar HDI to India, and perhaps to include another country.
- Instead the table 2, the authors could write about the trends in a similar way as the line 36 in the page 6. Perhaps to emphasize the trends for India, North region (where is Rajasthan, the place of the study), North East and South regions too, just to abbreviate and highlight the bigger differences among region and states
- In line 41 in the page 6, the authors wrote that the highest declines happened in states with better health systems, but, it could be important another information that could justify the differences among states
- Also, it could be important the authors to bring information about the Rajasthan state, specifically
- The line 45 to 50 in page 7, the authors could write a little more about the differences inside the Rajasthan state, instead the table 3
- The table 3 could be an Appendix because it does not correspond to the information in lines 45 to 50 in page 7
- With the suggestion above, the authors will bring a view of what is happening at the country, state and district level
- In page 9, line 17, what does POSHAN mean?
- The figures 1, 2 and 5 do not have title as the figures 3 and 4

Methods:

- In page 11, in the paragraph about “Respondent selection” it could be good to justify the number of respondents, if it was one for each village for each category of respondent?
- Still in “Respondent selection”, were there other criteria to include the mothers and grandmothers for the study?
- In page 11, for the “data collection”, were there a number of questions for each topic for the FGD?
- In page 12, for “data processing and analysis”, was used a specific software for qualitative data? And, perhaps it could be good to include a reference about triangulation

Results:

- In addition to the table 4, it could be interesting to know the number of respondents for each village, for example, how many mothers and grandmothers were selected for each village
- The first paragraph of results could include a brief description of the characteristics of the mothers and grandmothers which participated in the FGD, for example, age, educational level or occupation, number of children.
- In page 14, line 19, about the initiation of breastfeeding, the information from the grandmothers could have a memory bias. The authors could clarify if they were present at the birth
- In page 15, line 5, for the first time appears the initials “FHW”. The authors could write the meaning
- In page 16, Core Theme 3, in line 5 to 10, the text is referring to the “delay” for complementary feeding; and in line 17 to 24, the text is referring to “frequency”. Therefore, the authors could

	consider “delay” and “frequency” in the figure 3 respect to the core theme 3 because were more reported, even more than “home cooked meals”  • In page 19, about the core theme 6, is reporting that the monitoring of the nutritional status is weak. Perhaps the authors could consider in the figure 4, for the core theme 6, in the “inadequate health and nutrition services” to include the word “monitoring” and not just “services” because “services” could have an broad meaning, and according to the text in page 19 is been reporting about monitoring actions too • In page 20, about core theme 7, was reported that the Panchayat had a weal role in health and nutrition issues, so, perhaps the authors could consider to include in the figure 4, in the respective core theme, a box with “weak (or other word) role of the Panchayat” • In page 21, about core theme 8, in line 23 to 33, was reported that older people and children practice open defecation, so, the authors could consider this to include a box in the figure 4 in the corresponding core theme, referring to “defecation practices” because is an important cause of infections in villages specially when there is not an adequate practice of hand washing Discussion  • It could be helpful that the authors bring some limitations, perhaps about the collection of data, logistic, coordination, if were people which refused to participate in the study what were the reasons?, how the researcher encouraged the participation of mothers, grandmothers and all type of respondents
--	--

VERSION 1 – AUTHOR RESPONSE

Reviewer: 2

Dr. Maria del Pilar Flores-Quispe, Universidade Federal de Pelotas

Comments to the Author:

My comments are listed below and are made by manuscript section.

Introduction:

- In line 9, I suggest to specify that the information about stunting and wasting is for children under-five

Thank you for pointing it out – the relevant information has been added

- In line 14 and 15, according to the reference number 2, the information about India is for the year 2007?, It could be possible to bring information of recent years, as the authors have made for the line 9

This was updated thank you! The latest reference has been added as UNICEF-WHO-World Bank: Joint Child Malnutrition Estimates 2018. From <https://data.unicef.org/resources/levels-and-trends-in-child-malnutrition-2018/>

- In the fourth paragraph, it could be better to describe, in a brief way, the differences between India and the other countries in South Asia instead the table 1, because in the introduction the authors should present a general view what is happening about the issue.

Thank you. Based on this comment, changes have been made where difference between India and other countries in South Asia have been briefly described.

- I could suggest writing about the breastfeeding indicators and compare to Global and Sri Lanka data. Although, Sri Lanka is a country smaller than India and has Human Development Index (HDI) higher than India, so, the authors perhaps could compare with other countries with characteristics similar to India.

UNICEF-WHO-World Bank: Joint Child Malnutrition Estimates 2018. Available From: <https://data.unicef.org/resources/levels-and-trends-in-child-malnutrition-2018/>

- Regarding to complementary feeding indicators, the authors could write comparing India with global and Bangladesh data, which has a similar HDI to India, and perhaps to include another country.

Based on the above 2 comments, changes have made in the paper and Table 1 can be removed

- Instead the table 2, the authors could write about the trends in a similar way as the line 36 in the page 6. Perhaps to emphasize the trends for India, North region (where is Rajasthan, the place of the study), North East and South regions too, just to abbreviate and highlight the bigger differences among region and states

- In line 41 in the page 6, the authors wrote that the highest declines happened in states with better health systems, but, it could be important another information that could justify the differences among states

- Also, it could be important the authors to bring information about the Rajasthan state, specifically

- The line 45 to 50 in page 7, the authors could write a little more about the differences inside the Rajasthan state, instead the table 3

- The table 3 could be an Appendix because it does not correspond to the information in lines 45 to 50 in page 7

- With the suggestion above, the authors will bring a view of what is happening at the country, state and district level

Based on the above 6 comments, changes have made in the paper and both the Tables 2 and 3 can be removed, while a new Table can be added (given in the text)

- In page 9, line 17, what does POSHAN mean?

Appropriate changes made and abbreviation explained

- The figures 1, 2 and 5 do not have title as the figures 3 and 4

We added a title for figures 1, 2 and 5, thank you!

Methods:

- In page 11, in the paragraph about “Respondent selection” it could be good to justify the number of respondents, if it was one for each village for each category of respondent?

Thank you for the comment. We have added the number of respondents for each category and each village as requested.

- Still in “Respondent selection”, were there other criteria to include the mothers and grandmothers for the study?

Thank you for the comment. We have added two other criteria to include the mothers and the grandmothers for the study.

- In page 11, for the “data collection”, were there a number of questions for each topic for the FGD?

Thank you. We have added the topics and an approximate number of questions that were asked under each topic.

- In page 12, for “data processing and analysis”, was used a specific software for qualitative data? And, perhaps it could be good to include a reference about triangulation

Thank you for these comments. We have added the specific software that is used for the qualitative data analysis. We have also added reference about triangulation.

Results:

- In addition to the table 4, it could be interesting to know the number of respondents for each village, for example, how many mothers and grandmothers were selected for each village

Thank you very much for this comment. The total number of respondents for each village including mothers, grandmothers and key informants have been added at the beginning of this section before the Table 4.

- The first paragraph of results could include a brief description of the characteristics of the mothers and grandmothers which participated in the FGD, for example, age, educational level or occupation, number of children.

Thank you very much for the comment. We have added a brief description about the caregivers (mothers and grandmothers) in the beginning of the section, that includes information on their age, literacy, occupation and number of children.

- In page 14, line 19, about the initiation of breastfeeding, the information from the grandmothers could have a memory bias. The authors could clarify if they were present at the birth

Thank you for the comment. The information has been clarified by mentioning about the presence of the mothers at the time of birth.

- In page 15, line 5, for the first time appears the initials "FHW". The authors could write the meaning

Thank you. We have written the meaning of the initials FHW in page 15, line 5.

- In page 16, Core Theme 3, in line 5 to 10, the text is referring to the "delay" for complementary feeding; and in line 17 to 24, the text is referring to "frequency". Therefore, the authors could consider "delay" and "frequency" in the figure 3 respect to the core theme 3 because were more reported, even more than "home cooked meals"

Thank you for the comment. The 'Delay and frequency for complimentary feeding' has been added in figure 3.

- In page 19, about the core theme 6, is reporting that the monitoring of the nutritional status is weak. Perhaps the authors could consider in the figure 4, for the core theme 6, in the "inadequate health and nutrition services" to include the word "monitoring" and not just "services" because "services" could have an broad meaning, and according to the text in page 19 is been reporting about monitoring actions too

Thank you for the comment. The word 'monitoring' has been added in figure 4.

- In page 20, about core theme 7, was reported that the Panchayat had a weal role in health and

nutrition issues, so, perhaps the authors could consider to include in the figure 4, in the respective core theme, a box with “weak (or other word) role of the Panchayat”

Thank you for the comment. A box with a title ‘lack of awareness and governance in panchayat’ has been added in Figure 4.

• In page 21, about core theme 8, in line 23 to 33, was reported that older people and children practice open defecation, so, the authors could consider this to include a box in the figure 4 in the corresponding core theme, referring to “defecation practices” because is an important cause of infections in villages specially when there is not an adequate practice of hand washing

Thank you very much. A box is added with the title of Open defecation practices by children and older people in Figure 4.

Discussion

• It could be helpful that the authors bring some limitations, perhaps about the collection of data, logistic, coordination, if were people who refused to participate in the study what were the reasons? how the researcher encouraged the participation of mothers, grandmothers, and all type of respondents

Thank you very much for all these comments. We have added the limitation of the study with respect to logistics and data collection on page 23. We also mentioned about the respondents who refused to participate along with the reasons and what was done to encourage the participation of all of them.

VERSION 2 – REVIEW

REVIEWER	Mamidi, Raja Sriswan Indian Council of Medical Research, Clinical Epidemiology
REVIEW RETURNED	09-Jan-2022

GENERAL COMMENTS	General comment 1. Since NFHS-5 is out since review, the authors may consider adding it as another column of data, wherever present for time trends. Specific comments 1. COREX checklist 13 and 15 may be filled as NA and not kept blank 2. Figure 1, Vaccination seeking and Supervision, training heads have unclear arrows. need to make them visible.
---

REVIEWER	Flores-Quispe, Maria del Pilar Universidade Federal de Pelotas, Medicina Social
REVIEW RETURNED	25-Jan-2022

GENERAL COMMENTS	The manuscript “Why India is struggling to feed their young children? A Qualitative analysis for tribal communities” submitted to BMJ Open is a very interesting paper that brings a import issue about the mechanism that affect the IYCF practices for children under two years old, including three levels, individual, village and governance level, a relevant phase as the PANChSHEEEL project. The methodology was well described, and the mean result, the conceptual framework was well elaborated considering the findings. The authors answered the comments made in the previous reviewing.
---

VERSION 2 – AUTHOR RESPONSE

Reviewer: 1

Dr. Raja Sriswan Mamidi, Indian Council of Medical Research

Comments to the Author:

General comment

1. Since NFHS-5 is out since review, the authors may consider adding it as another column of data, wherever present for time trends. (add it to wherever relevant)

Thank you very much for this. Data from NFHS-5 have been added to Table 1 and wherever required in the research paper.

Specific comments

1. CORE X checklist 13 and 15 may be filled as NA and not kept blank

Thank you very much for this comment. This change has been incorporated.

2. Figure 1, Vaccination seeking and Supervision, training heads have unclear arrows. need to make them visible.

Thank you very much for pointing this out. The arrows to and from Vaccination seeking and Supervision & Training have been made clear.

Reviewer: 2

Dr. Maria del Pilar Flores-Quispe, Universidade Federal de Pelotas

Comments to the Author:

The manuscript “Why India is struggling to feed their young children? A Qualitative analysis for tribal communities” submitted to BMJ Open is a very interesting paper that brings a import issue about the mechanism that affect the IYCF practices for children under two years old, including three levels, individual, village and governance level, a relevant phase as the PANChSHEEEL project. The

methodology was well described, and the mean result, the conceptual framework was well elaborated considering the findings. The authors answered the comments made in the previous reviewing.

Thank you very much for your reviews and comments. It has been very helpful.

Reviewer: 1

Competing interests of Reviewer: None

Reviewer: 2

Competing interests of Reviewer: I have not competing interests

VERSION 3 – REVIEW

REVIEWER	Mamidi, Raja Sriswan Indian Council of Medical Research, Clinical Epidemiology
REVIEW RETURNED	29-Jun-2022

GENERAL COMMENTS	This is a fantastic piece of work by the authors and I recommend them to promote it during the POSHAN abhiyaan meeting conducted by Niti Agog, Gol.
---